# Computer-Aided Design and Computer-Aided Modeling (CAD/CAM) for Guiding Dental Implant Surgery: Personal Reflection Based on 10 Years of Real-Life Experience

**DOI:** 10.3390/jpm13010129

**Published:** 2023-01-09

**Authors:** Paolo Scolozzi, Francesco Michelini, Claude Crottaz, Alexandre Perez

**Affiliations:** 1Division of Oral and Maxillofacial Surgery, Department of Surgery, Faculty of Medicine, Hôpitaux Universitaire de Genève, 1211 Genève, Switzerland; 2Unit of Oral Surgery and Implantology, Division of Oral and Maxillofacial Surgery, Department of Surgery, Faculty of Medicine, Hôpitaux Universitaire de Genève, 1211 Genève, Switzerland

**Keywords:** computer-assisted surgery, virtual surgical planning, guided implant surgery, computer-aided design/computer-aided manufacturing software (CAD/CAM), surgical guide

## Abstract

Traditional dental implant surgery has been challenged by the phenomenal progression in computer-assisted surgery (CAS) that we have been witnessing in recent years. Among the computer-aided technologies, computer-aided design and computer-aided manufacturing (CAD/CAM) techniques represent by far the most attractive and accepted alternatives over their dynamic counterpart, navigational assistance. Based on many years of experience, we have determined that CAD/CAM technology for guiding dental implant surgery is valuable for rehabilitation of the anterior maxillary region and the management of complete or severe partial edentulism. The technology also guarantees the 3D parallelism of implants. The purpose of the present report is to describe indications for use of CAD/CAM dental implant guided surgery. We analyzed the clinical and radiological data of thirteen consecutive edentulous patients treated using CAD/CAM techniques. All of the patients had stable cosmetic results with a high rate of patient satisfaction at the final follow-up examination. No intra- and/or postoperative complications were encountered during any of the steps of the procedure. The application of CAD/CAM techniques produced successful outcomes in the patients presented in this series.

## 1. Introduction

Clinical esthetic evaluation, study models, surgical templates and plane radiographs (e.g., panoramic and retroalveolar views) are currently the tools most widely used by oral and maxillofacial surgeons worldwide in planning and performing traditional “freehand” implant surgery [1,2]. This procedure provides the advantages of being easy, adequate, rapid and cost-effective without having to resort to a constantly evolving, demanding and expensive technology. However, the accuracy of reproducing the traditional “hand-made” simulation during surgery has always been the major drawback of such a technique [1,2]. This is mainly due to inevitable cumulative errors related to the transfer of 2D plane X-rays and information from articulated plaster models in planning 3D implant positioning, especially in fully edentulous cases.

In the last two decades, the traditional method of implant surgery has been challenged by an unprecedented progression in new imaging technology and medical informatics, which paved the way for the future development of computer-assisted surgery (CAS) [3,4]. The CAS modalities, which have been traditionally separated into dynamic and static, aid the surgeon in the visualization and transfer of the preoperative 3D virtual surgical planning for the patient [5,6,7]. Dynamic CAS is performed via the use of navigation assistance, which allows for the simultaneous real-time 3D localization of tracked drilling instruments on the patient’s target area as well as on the preoperative images, thus magnifying the details provided by the real world [5,6,7]. Conversely, static CAS is driven by CAD/CAM technology, which enables the transformation of a virtual file into real objects, such as patient-specific surgical drilling guides. Such guides, which are tailored according to the individual’s anatomy, allow for the transfer of the ideal virtual planned implant’s position into surgery [5,6,7]. 

Undoubtedly, the CAD/CAM technology has become the preferred method over navigational assistance, establishing itself as a versatile and valid alternative to the traditional technique for the management of a complex implant’s supported prosthetic restorations [5,6,7]. Moreover, the spectacular progression of CAD/CAM technology in dental implant treatment that we have witnessed in recent years has been mainly fueled by the adoption of the “prosthetically driven implant placement” paradigm, which requires higher accuracy to achieve an ideal esthetic and functional outcome [8,9,10]. 

Although several studies have demonstrated the accuracy of CAS technology in dental implant surgery and have claimed the superiority of this approach over traditional techniques, to date, only a meta-analysis has been published reporting on the comparison of computer-assisted technology to the conventional freehand technique and found no significant differences between the two approaches for marginal bone loss, mechanical and biological complications and implant survival rate [10,11,12,13,14,15]. However, these results should be interpreted with caution given the lack of homogeneity that limited this systematic review. In light of this observation and in the absence of certainties, one should thus wonder in which situations CAS would be of real “added value” compared to conventional techniques. The answer must be sought in a comprehensive clinical assessment of biological, functional and esthetic success and not as a matter of mere geometrical accuracy.

The purpose of the present report is to describe the indications for the use of patient-specific computer-aided design and computer-aided modeling (CAD/CAM) for guided dental implant surgery based on our 10 years of real-life experience.

## 2. Patients and Methods

### Study Design

This retrospective study included all patients who had guided implant surgery using CAD/CAM Simplant^®^ software (Version 18.5, https://www.dentsplysirona.com) at the Hôpitaux Universitaires de Genève, Switzerland, between 2012 and 2021. 

The procedure followed in this retrospective study was in accordance with the Helsinki Declaration of 1975, as revised in 2000, and was approved by the Clinical Ethics Council (CEC) of the Hôpitaux Universitaires de Genève (No 2019-01714).

The variables reviewed included age and gender, type of edentulism, surgical procedure and postoperative complications.

## 3. Technical Protocol

CAD/CAM patient-specific surgical drilling guides were planned and constructed according to the following protocol:

### 3.1. Step 1: Planning of the Ideal Implant-Supported Prosthesis

The first prototype of the ideal prosthesis was created based on articulated plaster models obtained by conventional impressions. The prosthesis was designed to meet the best esthetic and functional requirements.

### 3.2. Step 2: CBCT Scan of the Patient and the Prosthesis with Integrated Reference Markers (Dual-Scan Technique)

A CBCT scan was performed with the patient wearing the planned prosthesis equipped with integrated radiopaque reference spheres. Then, a second scan of the prosthesis was taken, applying the same general settings that were used for obtaining the first scan. However, a higher resolution was used to allow a very accurate visualization of the teeth and the markers.

### 3.3. Step 3: Preoperative Virtual Planning in the Software

#### 3.3.1. Image Transfer and Processing 

The CBCT scan images of the cranio-facial skeleton and the prosthesis were processed using Simplant^®^ software (Version 18.5, https://www.dentsplysirona.com) in DICOM (Digital Imaging and Communications in Medicine) format. The reference spheres visible in both scanned data sets were virtually superimposed, allowing for the realignment and fusion of the prosthesis with the patient CBCT data. 

#### 3.3.2. Image Segmentation and Virtual Planning

A semi-automatic segmentation of the volumetric region of interest of the prosthesis was performed on 3D images windowed into bone-specific Hounsfield units using a specific cursor. The number, the length, the diameter and the placement of computational virtual implants were determined with respect to the design of the final prosthesis, the available residual bone volume and the anatomical limitations (maxillary sinus, nasal floor, mandibular canal, etc.). 

### 3.4. Step 4: Manufacturing of the Surgical Stereolithographic Drilling Guide 

After the surgeon’s approval of the treatment plan, data were used to create the specific surgical drilling guides according to the number and placement of the implants previously determined based upon the images. The final nonsterile drilling guide together with the implant-guided procedure guideline was thus sent by the manufacturer to the surgeon and sterilized by autoclave prior to its utilization at the Hôpitaux Universitaires de Genève. 

## 4. Surgical Procedure

Prior to starting the drilling procedure, bone-supported and tooth-supported guides were temporarily stabilized by using two mini screws on the right and left sides or tooth anchorage, respectively. The stability of fit was then manually verified. The implant bed preparation was performed by strictly following the manufacturer’s guideline with regard to the use of specific ad hoc drill handles (T-sleeve position (H2, H4 or H6), cylinder heights (1 mm or 3 mm) and drill length (short, medium or long); (Straumann^®^ Guided Surgery System Instruments; www.straumann.com)) according to the initial planning. The fine implant bed preparation was then finished with profile drilling and tapping. The guided implant placement was realized by using visual depth control, and the insertion was finished by manual tightening of the implant in order to control the positioning approximately 2–3 mm apical to the alveolar crest or to the line joining the enamel–cement limit of the adjacent teeth. The drilling guide was removed, and the stability of the implants was tested by using the torque control device.

All the implants (Straumann^®^ Bone Level or Bone Level Tapered Implant; www.straumann.com) were submerged and incisions were closed with uninterrupted 3-0 Vicryl sutures (Ethicon Sàrl, Neuchâtel, Switzerland). The re-opening and prosthetic phase began 2 to 3 months postoperatively.

## 5. Results

The study sample included 13 patients (10 women and 3 men) and their ages ranged from 19 to 74 years (mean age: 46.2 years). The demographic, clinical, treatment and outcome characteristics of our sample are summarized in Table 1. Of the 13 patients, three (23%) had completely edentulous maxilla and mandible, two (15%) had partially edentulous maxilla and mandible, two (15%) had partially edentulous maxilla, two (15%) had partially edentulous mandible, two (15%) had partially edentulous anterior maxilla, one (8%) had completely edentulous mandible and one (8%) had completely edentulous mandible. Eight (61%) patients had pre-prosthetic surgical procedures as follows: six patients had bone graft augmentation (three had Le Fort I osteotomy with interpositional iliac crest graft, two had onlay iliac crest graft and one had a free fibular graft) and two patients had a bilateral inferior alveolar nerve lateral transposition. Bone-supported and tooth-supported guides were used in 10 and 3 patients, respectively. Ten (77%) patients had a fixed screw-retained prosthesis, and five (23%) had a bar-retained overdenture prosthesis.

No intraoperative technical and/or instrumentation complications were encountered during the different steps of the procedure. None of the patients required revision surgery. The follow-up period ranged from a minimum of 1 year to a maximum of 5 years (average 2.8 years), and the follow-up examination showed stable cosmetic and dimensional results in all patients. 

## 6. Illustrative Cases

### 6.1. Case 1 

#### 6.1.1. Anterior Maxillary Post-Traumatic Alveolar Ridge Defect (Maxillary Screw-Retained Implant-Supported Fixed Bridge)

A 30-year-old man was admitted to the Emergency Room at the Hôpitaux Universitaires de Genève in December 2014 following a ski accident. He experienced Le Fort II and trifocal mandibular fractures (right and left subcondylar and symphyseal region), which were treated by an immediate open reduction and internal rigid fixation with titanium miniplates in our department. He also had compound coronal fractures of the right lateral and central incisors that had to be removed as well as avulsion of the left central incisor. His postoperative course was followed by the loss of the right lateral and central incisors and the left central incisor. 

#### 6.1.2. Post-Bone-Grafted Documentation

In May 2015, he underwent onlay iliac crest bone graft augmentation for the residual maxillary alveolar ridge defect (Figure 1 and Figure 2).

Three-dimensional (3D) CT scan depicting the grafted maxilla (bilateral mandibular ramus cortical bone grafts with screw fixation).

#### 6.1.3. Image Segmentation and Virtual Planning: 

The number, the length, the diameter and the placement of computational virtual implants were determined with respect to the design of the final prosthesis the available residual bone volume and the anatomical limitations according to the previously described technical procedure (Figure 3, Figure 4, Figure 5 and Figure 6).

#### 6.1.4. Intra-Operative Documentation: 

In November 2017, two guided dental implants were placed at sites # 12 and # 21 (Figure 7, Figure 8, Figure 9, Figure 10 and Figure 11).

#### 6.1.5. Postoperative (5-Years Follow-Up) Documentation: 

A screw-retained provisional fixed prosthesis was used for 4 months to condition peri-implant mucosa, and in January 2019, the final prosthesis was installed. The occlusion was adjusted, and the patient received instruction for oral hygiene. A follow-up assessment at 5 years showed a stable cosmetic, biological and functional reconstruction (Figure 12, Figure 13 and Figure 14).

### 6.2. Case 2 

#### 6.2.1. Severe Partial Edentulous Maxilla and Complete Edentulous Mandible (Maxillary and Mandibular Screw-Retained Implant-Supported Bridges)

A 17-year-old girl was referred to our department for the management of a severe partial maxillary (persistence only of the central incisors # 11 and # 21) and complete mandibular edentulism following amelogenesis imperfecta with renal disease. In October 2011, she had the extraction of 20 impacted teeth. 

#### 6.2.2. Preoperative Documentation

In October 2012, she underwent multiple onlay iliac crest bone graft augmentation for the residual maxillary and mandibular alveolar ridge defect (Figure 15).

#### 6.2.3. Image Segmentation and Virtual Planning: 

The number, the length, the diameter and the placement of computational virtual implants were determined with respect to the design of the final prosthesis the available residual bone volume and the anatomical limitations according to the previously described technical procedure (Figure 16, Figure 17, Figure 18, Figure 19, Figure 20 and Figure 21).

#### 6.2.4. Intra-Operative Documentation: 

In March 2013, twelve guided dental implants were placed at sites # 12, # 13, # 15, # 22, # 23, # 25, # 33, # 34, # 36, # 43, # 44, and # 46 (Figure 22, Figure 23, Figure 24, Figure 25 and Figure 26)

#### 6.2.5. Postoperative (2-Years Follow-Up) Documentation: 

A screw-retained provisional fixed prosthesis was used for 4 months to condition peri-implant mucosa, and in October 2013, the final prosthesis was installed. The occlusion was adjusted, and the patient received instruction for oral hygiene. A follow-up assessment at 8 years showed a stable cosmetic, biological and functional reconstruction (Figure 27, Figure 28 and Figure 29).

### 6.3. Case 3 

#### 6.3.1. Complete Mandibular Edentulism (Mandibular Bar-Retained Implant-Supported Overdenture Using Straumann Pro Arch^®^ Protocol) 

A 72-year-old man was referred to our department for the investigation and management of a cheek squamous cell carcinoma in June 2017. He underwent tumor resection, which also involved partial mandibular coronoid process removal, neck dissection and immediate reconstruction of the cheek defect by a temporalis muscle flap. The postoperative course was uneventful, and follow-up at 1 year showed a stable situation free of any tumor relapse. 

#### 6.3.2. Preoperative Documentation: 

However, modification of the residual alveolar ridge due to the reconstructive surgery made the wearing of the previous conventional complete denture impossible. Therefore, rehabilitation using an implant-supported overdenture was planned (Figure 30).

#### 6.3.3. Image Segmentation and Virtual Planning: 

The number, the length, the diameter and the placement of computational virtual implants were determined with respect to the design of the final prosthesis the available residual bone volume and the anatomical limitations according to the previously described technical procedure (Figure 31, Figure 32, Figure 33 and Figure 34).

#### 6.3.4. Intra-Operative Documentation: 

In August 2019, four guided dental implants were placed at sites # 34, # 31, # 41, and # 44 (Figure 35 and Figure 36).

#### 6.3.5. Postoperative (3-Years Follow-Up) Documentation: 

In March 2020, an implant-supported overdenture with CAD/CAM bar attachment was placed. The occlusion was adjusted, and the patient received instruction for oral hygiene. A follow-up assessment at 3 years showed a stable cosmetic, biological and functional reconstruction (Figure 37 and Figure 38).

## 7. Discussion

The present study has summarized the current application of CAD/CAM technology in dental implant treatment used at the Hôpitaux Universitaires de Genève for the rehabilitation of anterior maxillary partial defects as well as for the rehabilitation of severe partial or complete edentulism. As a corollary, it should be mentioned that CAD/CAM software is routinely used for diagnostic purposes in all of our patients to visualize and determine the actual condition in detail. Thereafter, the simulation phase involves the manual placement of virtual dental implants to meet the requirements dictated by the design of the final prosthesis. This step is of paramount importance as it allows assessment of the adequacy of available bone and the possible anatomical limitations (maxillary sinus, nasal floor, mandibular canal, mental foramen, etc.) for implant placement in the desired locations. Finally, the decision to go through the guided phase by means of a stereolithographic drilling guide is exclusively dictated by the final prosthetic project. The virtual implant planning workflow implemented in our service combines the conventional prosthetic set-up aligned and registered with CBCT images as previously outlined. 

The CAD/CAM stereolithographic-guided implant placement has been associated with higher accuracy and reliability over the nonguided “freehand” technique, especially for implant placement in rehabilitation of the partially edentulous anterior maxilla, which is considered by several authors as the primary indication for computer-guided surgery in dental implantology [4,5,9].

### 7.1. Anterior Partial Maxillary Edentulism:

The surgical planning of implant placement to re-establish an esthetic maxillary incisor display requires perfect 3D control of: (1) parallelism between the implants; (2) mesio-distal position of the implants related to the adjacent teeth; (3) bucco-palatal position relative to the thickness of the alveolar ridge (1 mm palatal to the point of the emergence of the adjacent teeth); and (4) vertical position of the implants (2–3 mm apical relative to the zenith of the future prosthesis and the line joining the enamel-cement limit of the adjacent teeth). These are undeniably the most challenging and most crucial issues to deal with, which determine the successful clinical restoration of an esthetically balanced lip–incisor–gingival relationship [16]. In fact, fulfilling all of the above-mentioned requirements is essential for the realization of an occlusal screw-retained prosthesis, which remains the ideal esthetic solution in the anterior region. The advantages offered by this solution over the cement-retained prosthesis are predictable retrievability, soft tissue conditioning and finalization of the emergence profile during the provisional phase, and no risk of cement remnant, which can negatively influence peri-implant tissue health and potentially lead to the development of peri-implantitis and/or mucositis. The main disadvantage is related to the need for strict observance of the axis of implants in an ideal prosthetic position [16]. On the other hand, a cement-retained prosthesis does allow for axis rectification with CAD/CAM abutments, but this entails further non-negligible costs. Moreover, these restorations have been shown to be associated with increased risk of biological and technical complications (fracture/chipping of ceramic, loosening of abutment, fistula/suppuration and bone loss) [16]. Although these steps can be accurately managed only by 3D simulation in a virtual environment, the use of CAD/CAM guides should always be weighed against alternative positioning aids such as thermoformed guides. 

Thermoformed guides do not provide the same precision as the CAD/CAM guides, but they allow for better assessment and integration of the actual clinical parameters and for more flexibility by allowing the possibility of adapting the planning during the implant’s insertion [17]. This could be of great interest in particular cases, such as bone volume changes that could occur between the CBCT acquisition and the surgery; metal radiological artifacts related to restorations of the adjacent teeth, which can alter the correct assessment of the alveolar bone profile; inaccurate management of the threshold for the gray value of CBCT segmentation; or patient movement during the CBCT acquisition. In addition, thermoformed transparent guides offer other practical advantages including visualization of the real position of the tooth neck (dentin–enamel junction) at the zenith of the future prosthetic tooth, which determines the vertical position of the implant neck, and the evaluation of possible apical defect of the alveolar crest, which can entail the risk of a bone dehiscence during the surgical procedure leading to a more buccal orientation of the drilling axis to avoid the need for a bone graft procedure [16]. 

### 7.2. Severe Partial or Complete Edentulism:

The management of a full-arch prosthesis represents another critical situation that could greatly benefit from 3D virtual planning. In fact, the parallel positioning in all three planes of the spaces of the implants by using the traditional “freehand” technique is completely dependent on the 3D visualization capacity of the surgeon given that, in completely edentulous cases, there is no possibility to use conventional thermoformed guides. In this regard, the CAD/CAM technique provides several advantages: (1) visualization of the ideal site for implant placement to realize the ideal prosthetic, either fixed or removable; (2) manipulation of virtual implants, which allows choices about both the ideal size and perfect 3D spatial positioning with regard to the surrounding specific important anatomic structures, thus simplifying prosthetic project realization by eliminating the need for axis rectification with CAD/CAM abutments, which increases costs; (3) planning of the exact angulation of the tilted distal implants to avoid the maxillary sinus and/or the inferior alveolar nerve (e.g., All-on-4^®^ or Pro Arch^®^ protocol); (4) visualization and planning of the prosthetic space in the vertical plane (interocclusal space) needed to accommodate the implant attachment system (locator type attachment or a milled bar); and (5) accurate transfer of the 3D surgical planning to the patient in the operating room without having to constantly check the drilling axes during the surgical procedure, thus reducing the operative time to approximately 60 min, which represents the time spent for the online session planning. 

The main advantage related to the CAD/CAM approach is that the practitioner can accomplish the entire workflow process from surgical planning to final result without the need for highly specialized and trained staff or specific expensive infrastructure, which is in contrast to its navigation counterpart. However, on the other side, the CAD/CAM approach involves investing in continuously evolving dedicated software and implies a learning curve as well as a great deal of experience in conventional implantology. Moreover, by using this technique, the surgical treatment plan is rigidly locked with no possibility of real-time adaptation in case of unexpected deviations from the initial planning. Such an occurrence would require the surgeon to convert to the conventional technique during the surgical procedure. This would imply the consideration of a new prosthetic treatment plan during the surgery, relying exclusively on the surgeon’s ability to mentally represent the prosthetic project and the CBCT images simultaneously. This, although not impossible, requires significant experience, creativity and also a bit of luck.

Although, until recently, the high cost strongly limited the routine use of CAD/CAM-guided surgery, current prices have dropped significantly. Finally, it should also be noted that this technique is associated with cumulative technical and human errors related to imaging acquisition, software procedure of 3D segmentation, acquisition transformation of data into a guide and improper positioning of the guide during surgery that can impact the final outcome. According to the literature, intrinsic errors related to imaging acquisition processing and segmentation have been reported to be within 0.4 mm. 

## 8. Conclusions

So far, in our experience, computer-aided design and computer-aided modeling (CAD/CAM) for guiding dental implant surgery has been particularly suitable in the rehabilitation of anterior maxillary alveolar ridge defect and severe partial and complete maxillo-mandibular edentulism, for which traditional “freehand” management continues to be one of the most difficult and challenging procedures for oral and maxillofacial surgeons. Moreover, we consider the application of the present technique to be a valuable and successful alternative for the patients presented in this series. 

## Figures and Tables

**Figure 1 jpm-13-00129-f001:**
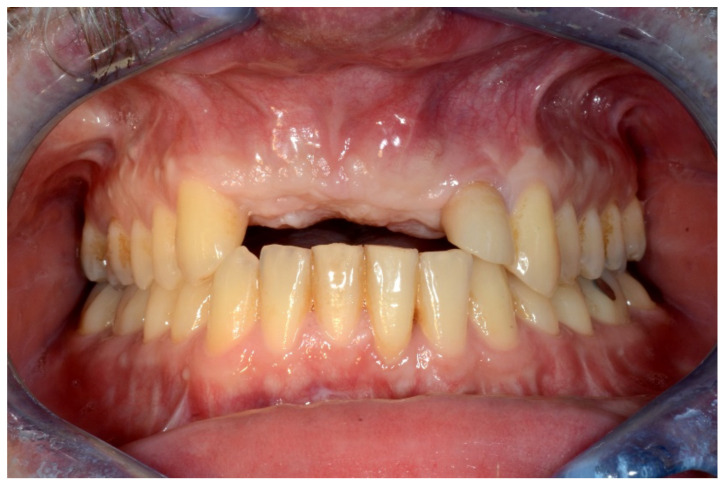
Frontal view showing the edentulous defect.

**Figure 2 jpm-13-00129-f002:**
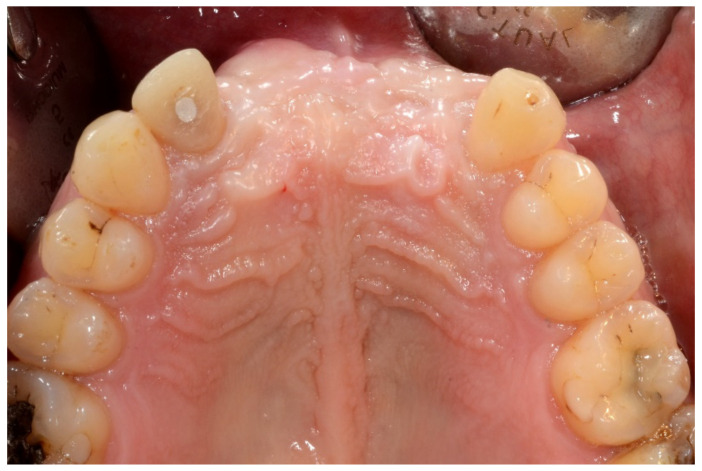
Occlusal intraoral view showing the edentulous defect.

**Figure 3 jpm-13-00129-f003:**
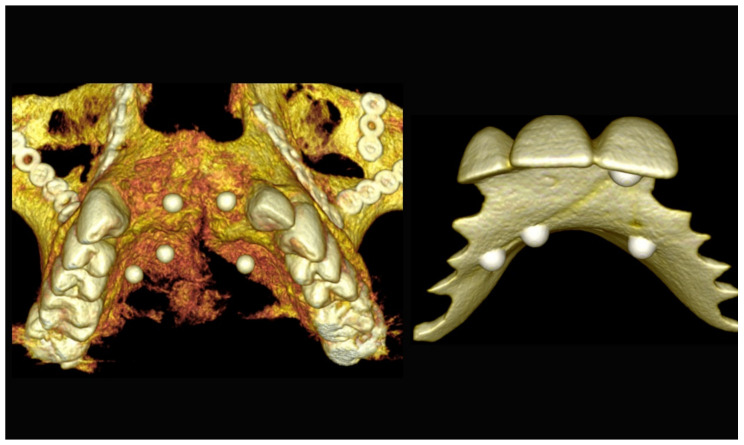
Three-dimensional (3D) CBCT scan images with the patient wearing the planned prosthesis and 3D images of the prosthesis equipped with integrated radiopaque reference spheres scanned separately.

**Figure 4 jpm-13-00129-f004:**
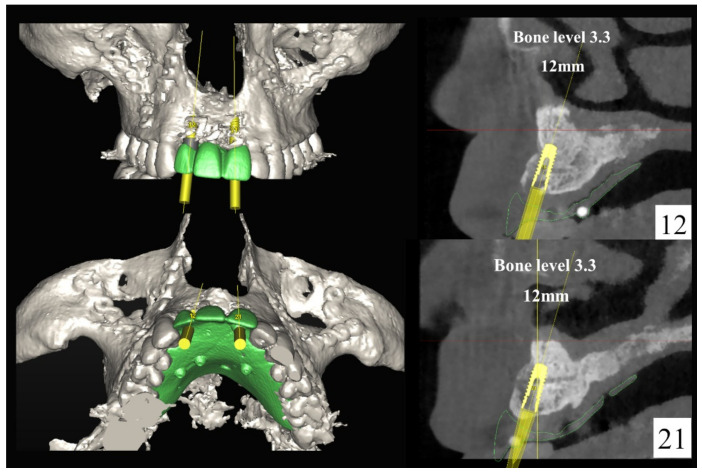
Planning of the number, the diameter, and the axis of the implant with respect to the design of the final prosthesis.

**Figure 5 jpm-13-00129-f005:**
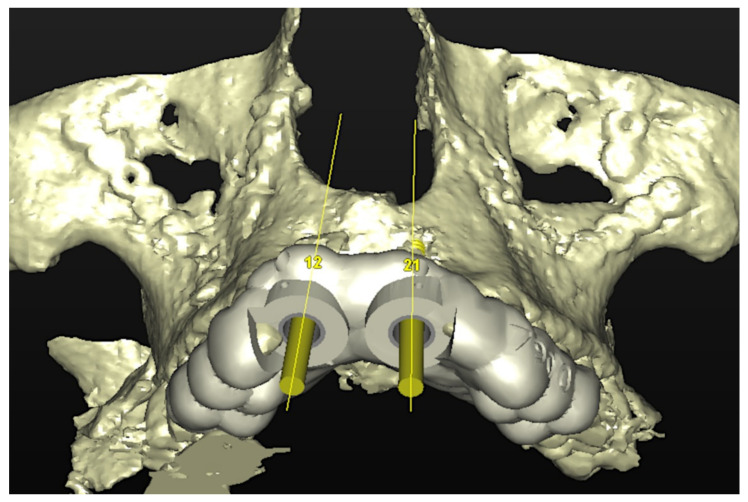
Virtual stereolithographic surgical drilling guides designed according to the implant position.

**Figure 6 jpm-13-00129-f006:**
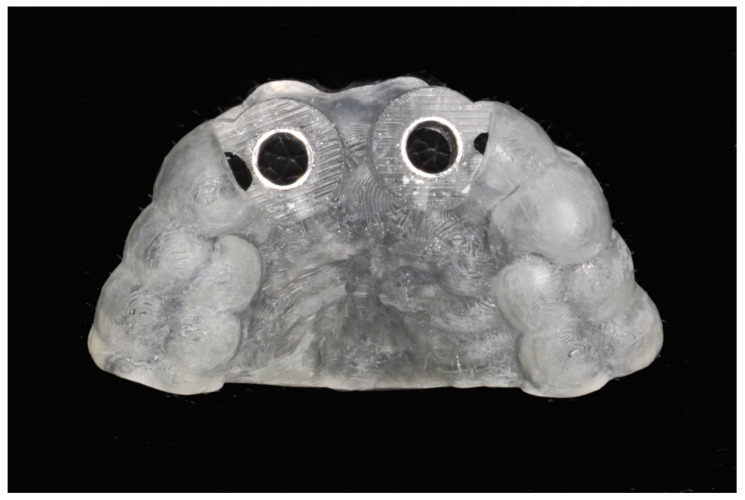
Final stereolithographic surgical drilling guides.

**Figure 7 jpm-13-00129-f007:**
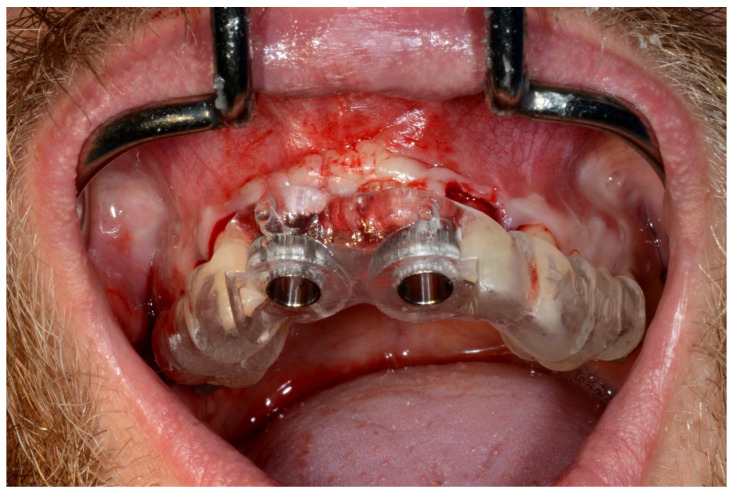
Surgical drilling guides temporarily supported by teeth on the right and left sides.

**Figure 8 jpm-13-00129-f008:**
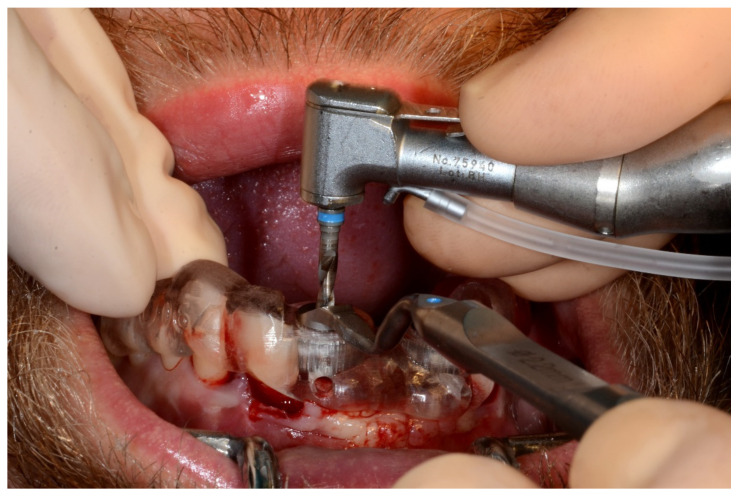
Implant bed preparation by using guided drills inserted into specific ad hoc drill handles.

**Figure 9 jpm-13-00129-f009:**
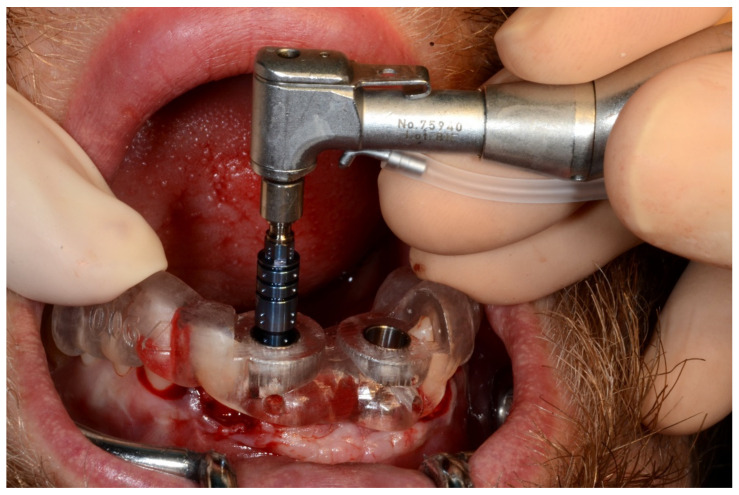
Guided implant insertion.

**Figure 10 jpm-13-00129-f010:**
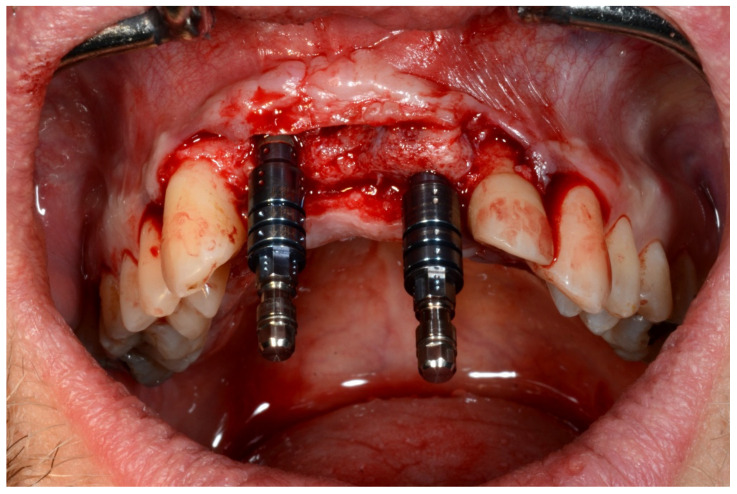
Frontal verification of the parallel placement of the implants.

**Figure 11 jpm-13-00129-f011:**
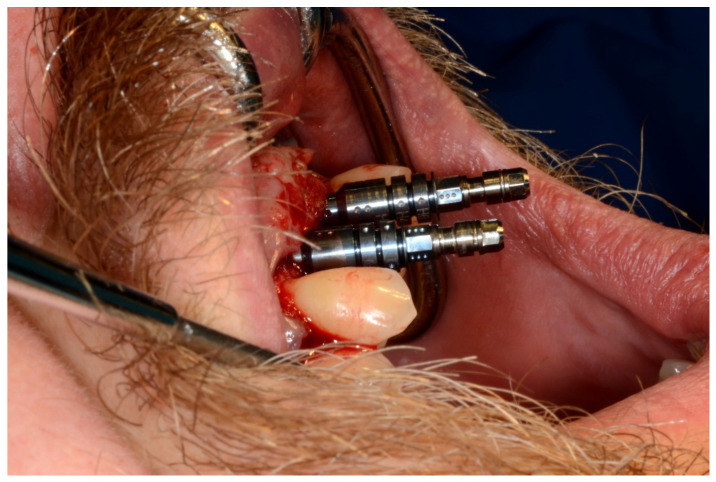
Lateral verification of the parallel placement of the implants.

**Figure 12 jpm-13-00129-f012:**
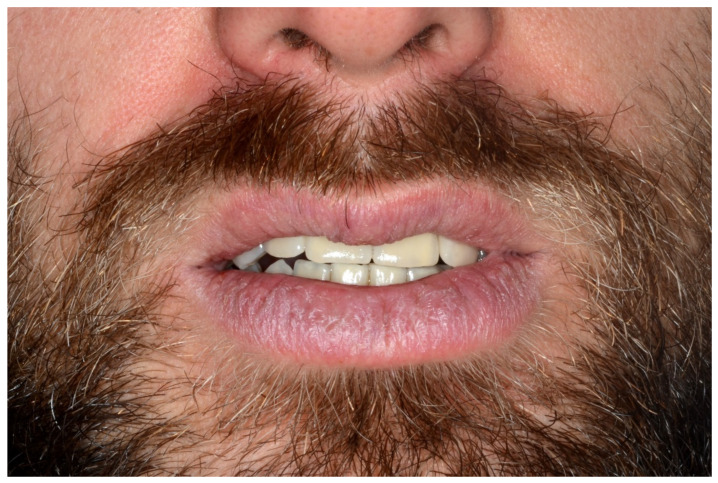
Resting frontal close-up views.

**Figure 13 jpm-13-00129-f013:**
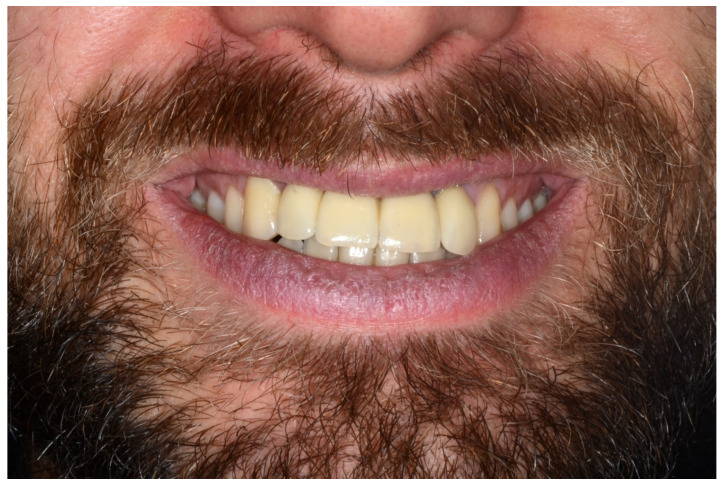
Smile frontal close-up views.

**Figure 14 jpm-13-00129-f014:**
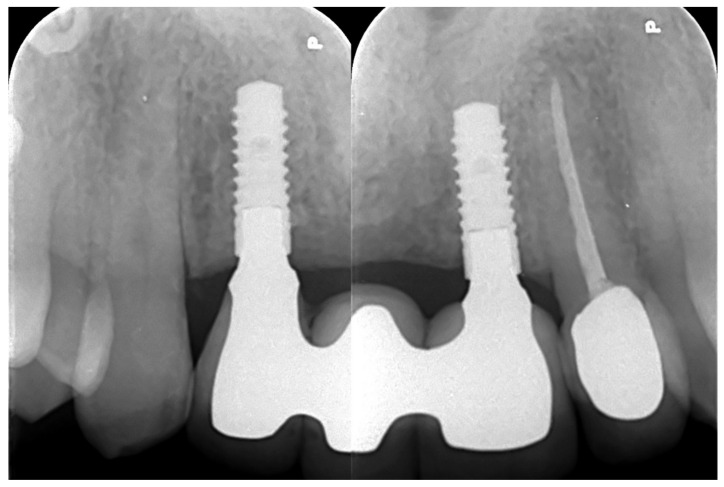
Retroalveolar radiographs showing the dental implants in place.

**Figure 15 jpm-13-00129-f015:**
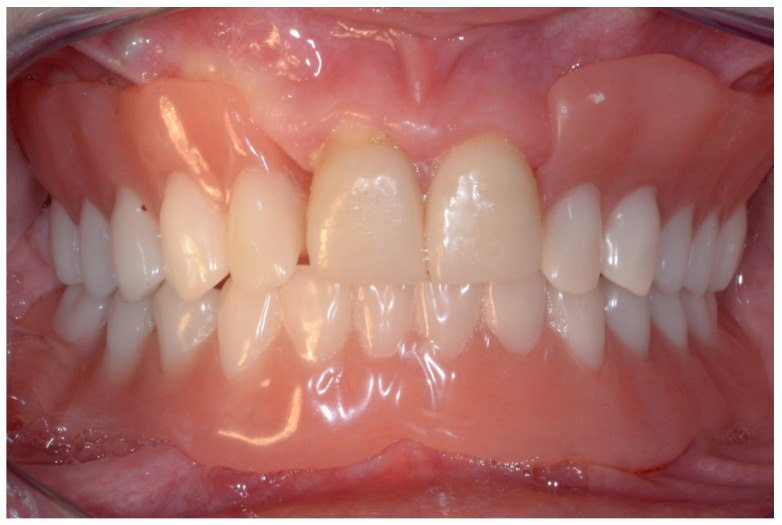
Maxillary and mandibular provisional prosthesis.

**Figure 16 jpm-13-00129-f016:**
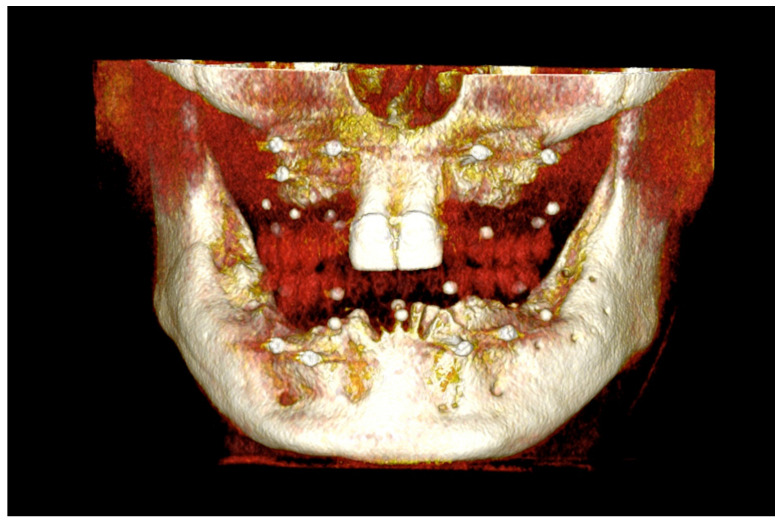
Three-dimensional (3D) CBCT scan images with the patient wearing the planned prosthesis.

**Figure 17 jpm-13-00129-f017:**
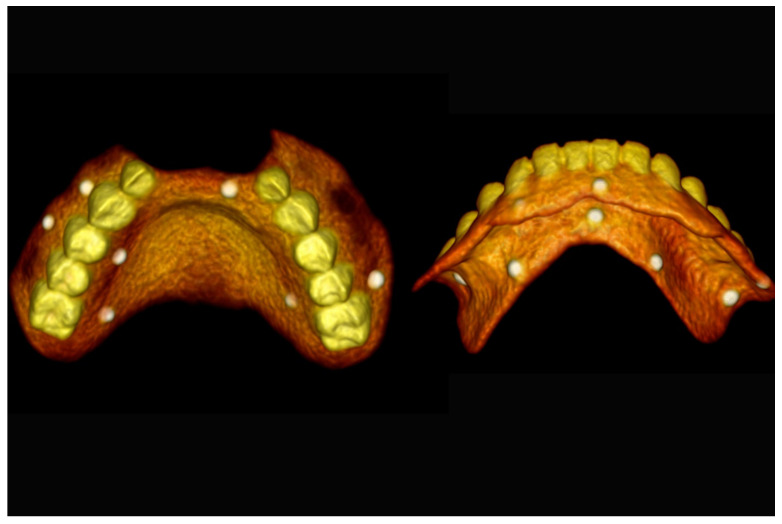
Three-dimensional (3D) CBCT scan images of the prosthesis equipped with integrated radiopaque reference spheres scanned separately.

**Figure 18 jpm-13-00129-f018:**
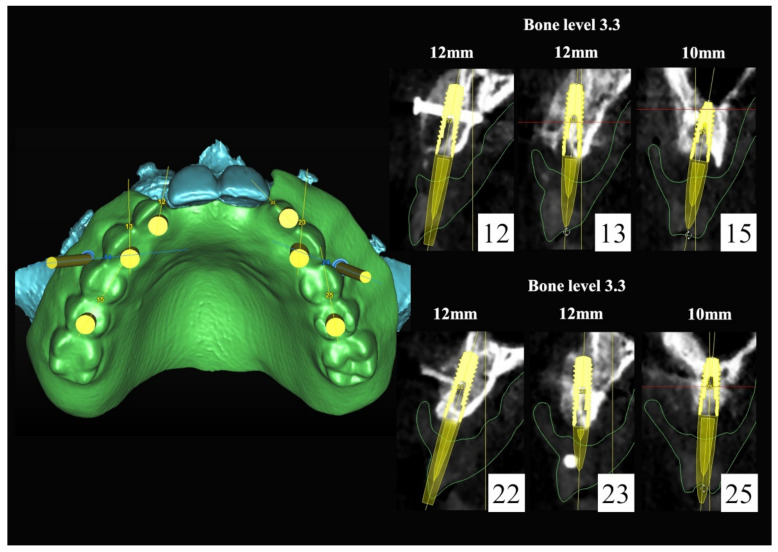
Three-dimensional (3D) planning of the number, the diameter and the axis of dental implants with respect to the design of the final virtual maxillary prosthesis.

**Figure 19 jpm-13-00129-f019:**
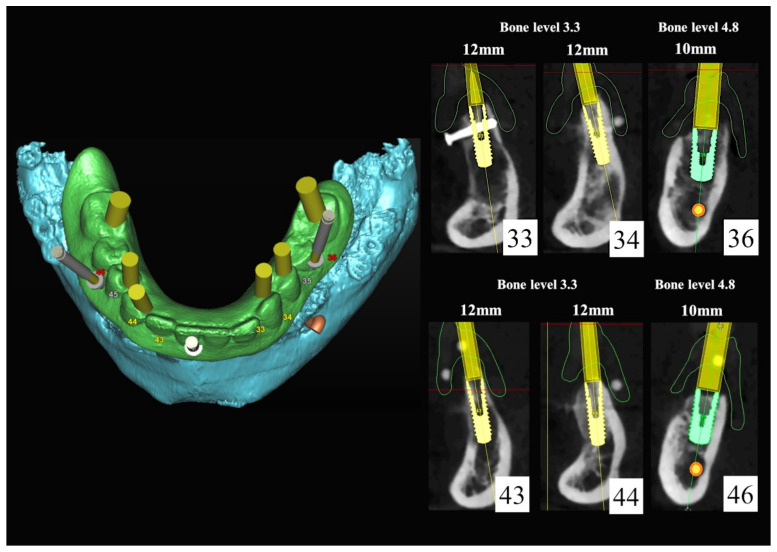
Three-dimensional (3D) planning of the number, the diameter and the axis of dental implants with respect to the design of the final virtual maxillary prosthesis.

**Figure 20 jpm-13-00129-f020:**
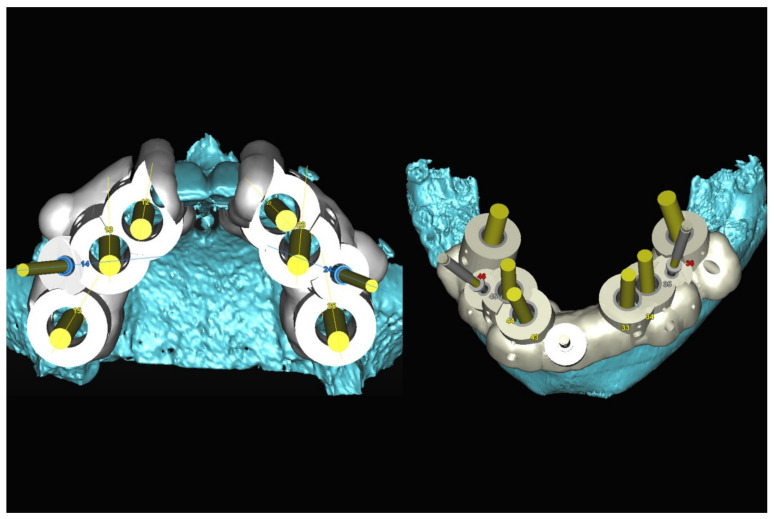
Maxillary and mandibular virtual surgical drilling guides designed according to the implant position.

**Figure 21 jpm-13-00129-f021:**
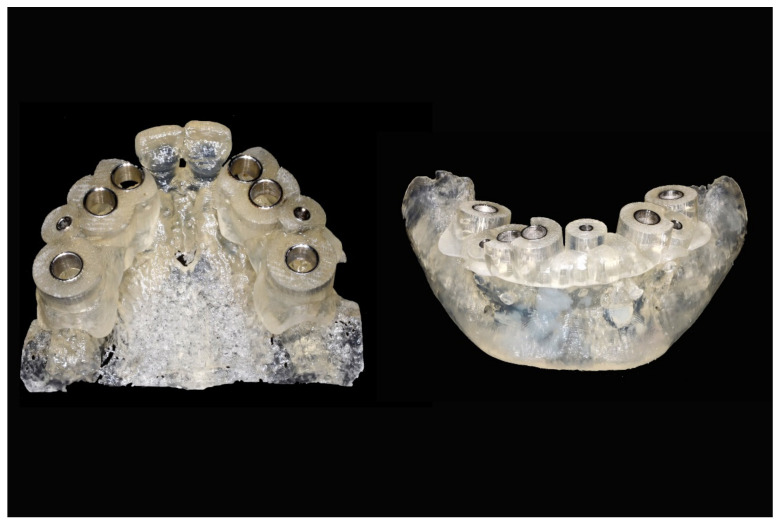
Final maxillary and mandibular stereolithographic surgical drilling guides.

**Figure 22 jpm-13-00129-f022:**
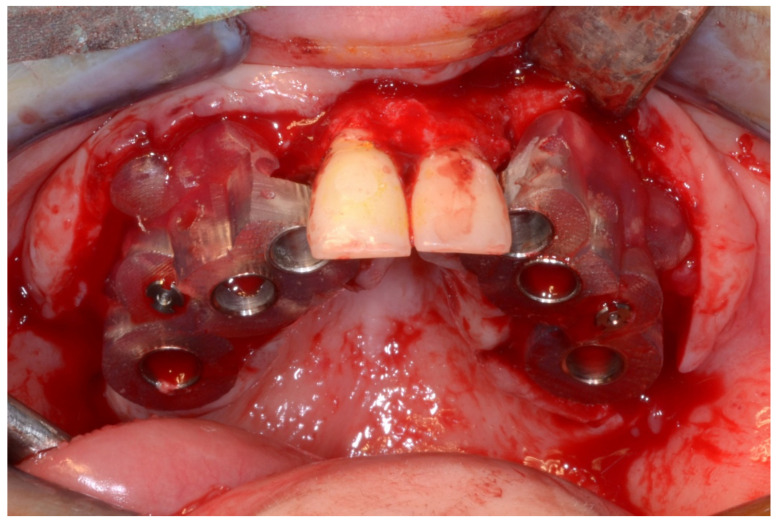
Maxillary surgical drilling guides temporarily supported and fixed with mini-screws.

**Figure 23 jpm-13-00129-f023:**
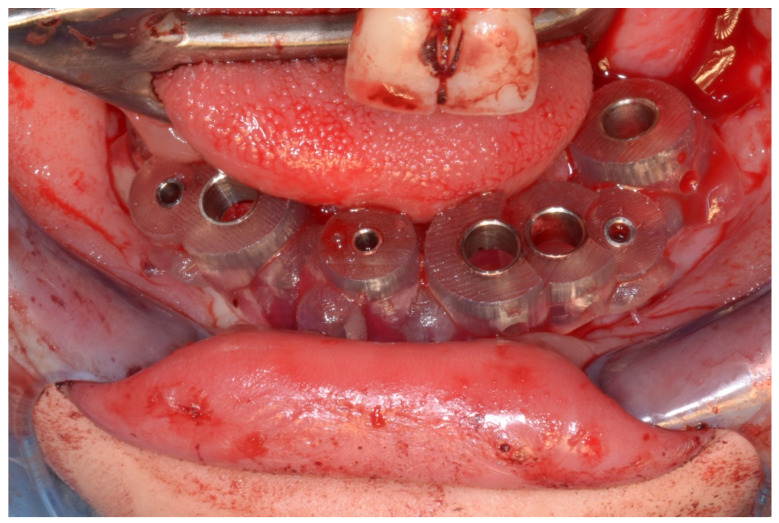
Mandibular surgical drilling guides temporarily supported and fixed with mini-screws.

**Figure 24 jpm-13-00129-f024:**
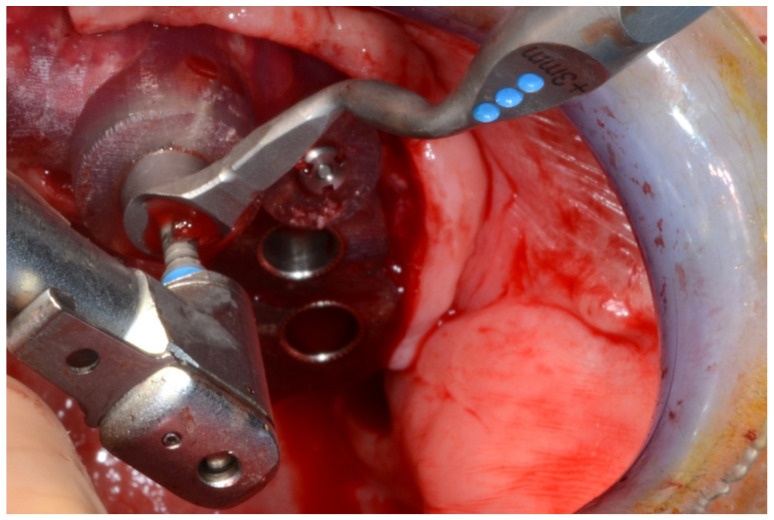
Implant bed preparation using guided drills inserted into specific ad hoc drill handles.

**Figure 25 jpm-13-00129-f025:**
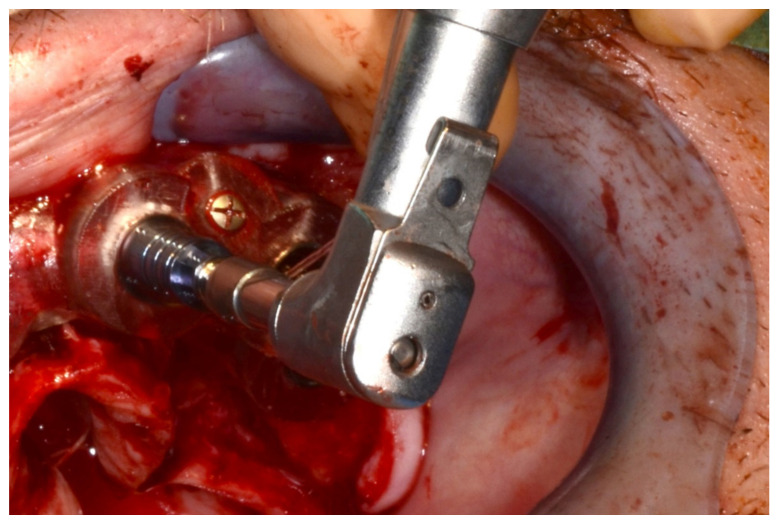
Guided implant insertion.

**Figure 26 jpm-13-00129-f026:**
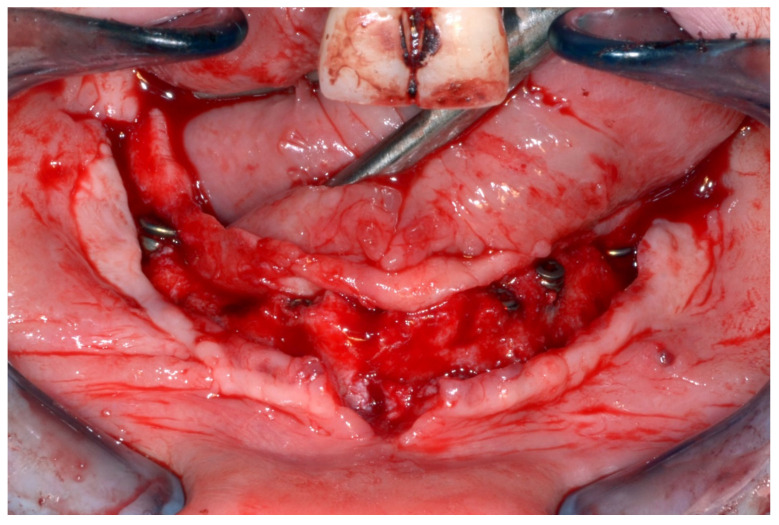
Mandibular implants in place.

**Figure 27 jpm-13-00129-f027:**
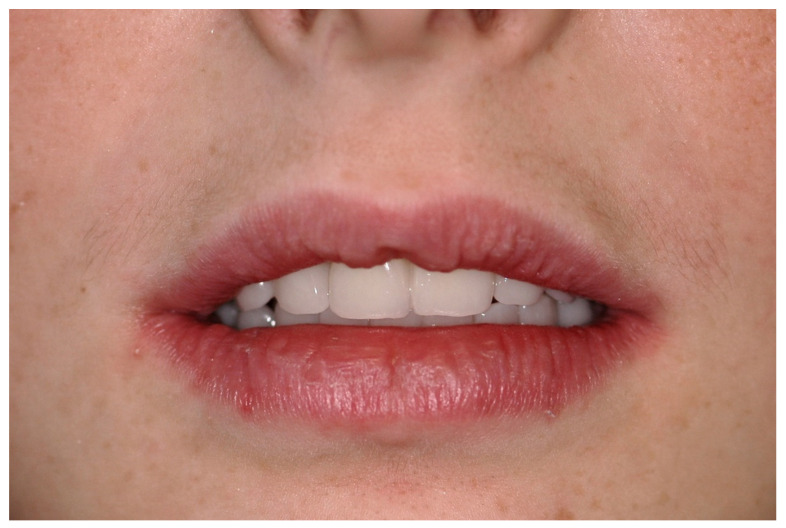
Resting frontal close-up views.

**Figure 28 jpm-13-00129-f028:**
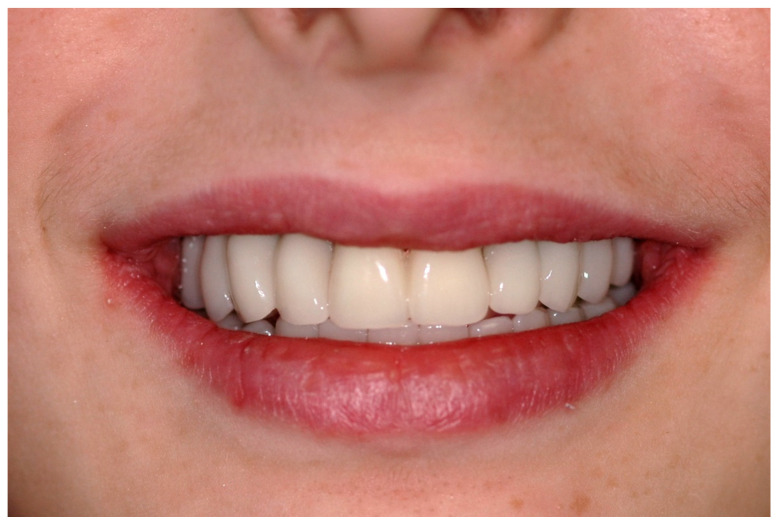
Smile frontal close-up views.

**Figure 29 jpm-13-00129-f029:**
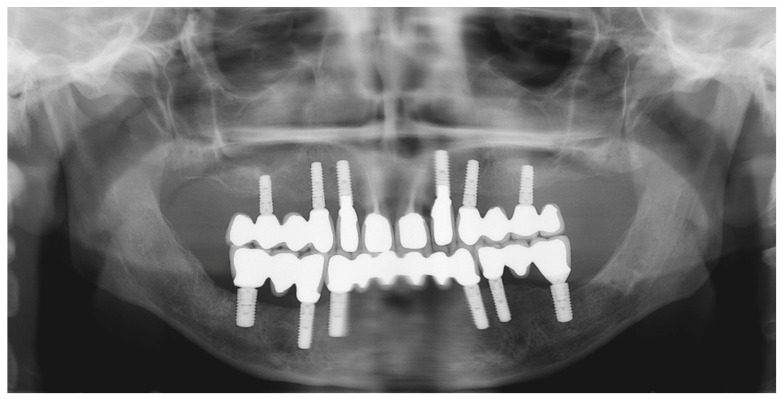
Panoramic radiograph showing the dental implants in place.

**Figure 30 jpm-13-00129-f030:**
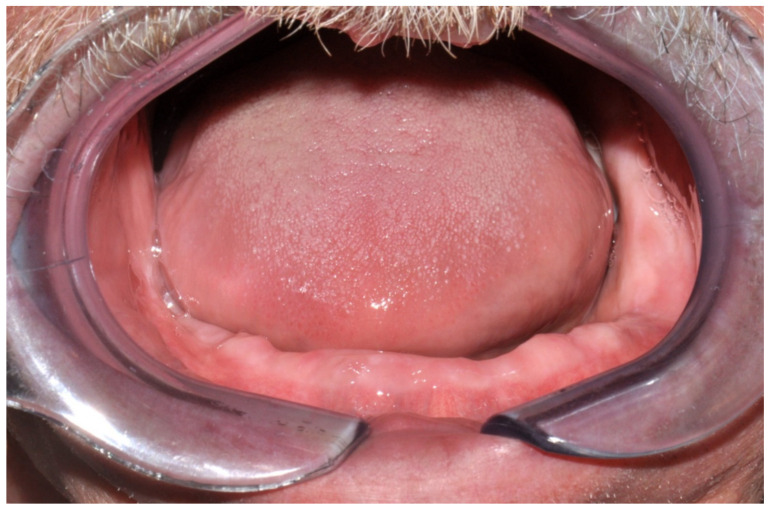
Intraoral view showing the edentulous defect.

**Figure 31 jpm-13-00129-f031:**
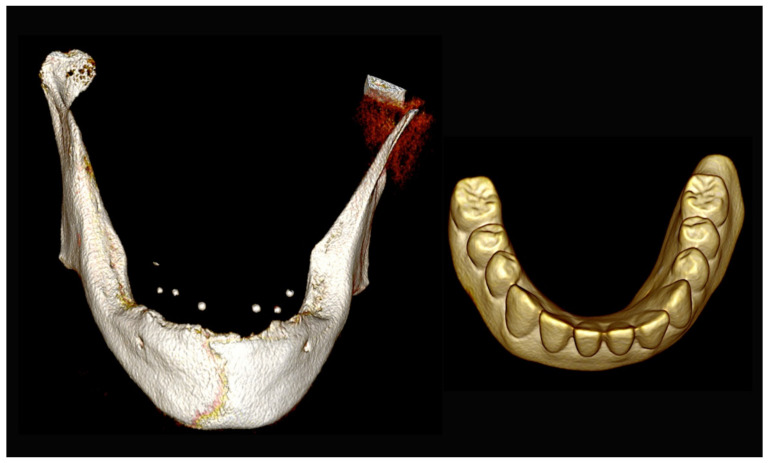
Three-dimensional (3D) CBCT scan images of the edentulous mandible with the patient wearing the provisional prosthesis and 3D images of the prosthesis equipped with integrated radiopaque reference spheres scanned separately.

**Figure 32 jpm-13-00129-f032:**
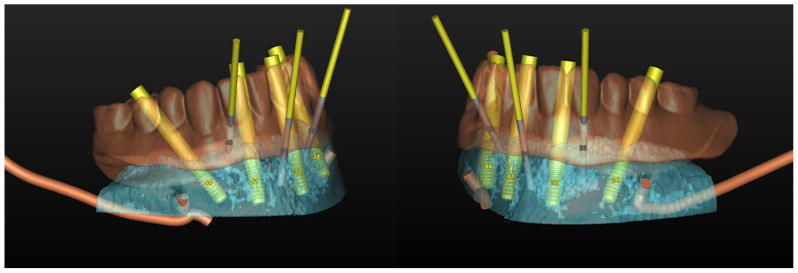
Planning of the number, the diameter, the axis of the implants with respect to the design of the final prosthesis and the trajectory of the inferior alveolar nerves.

**Figure 33 jpm-13-00129-f033:**
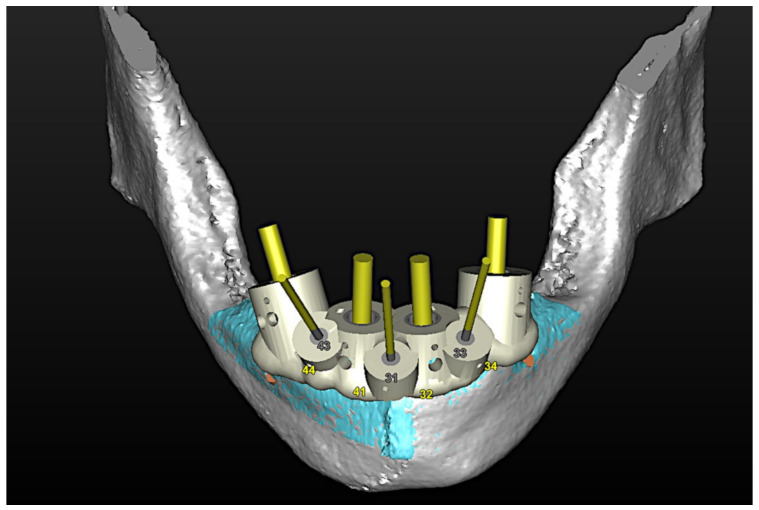
Virtual stereolithographic surgical drilling guides designed according to the implant position.

**Figure 34 jpm-13-00129-f034:**
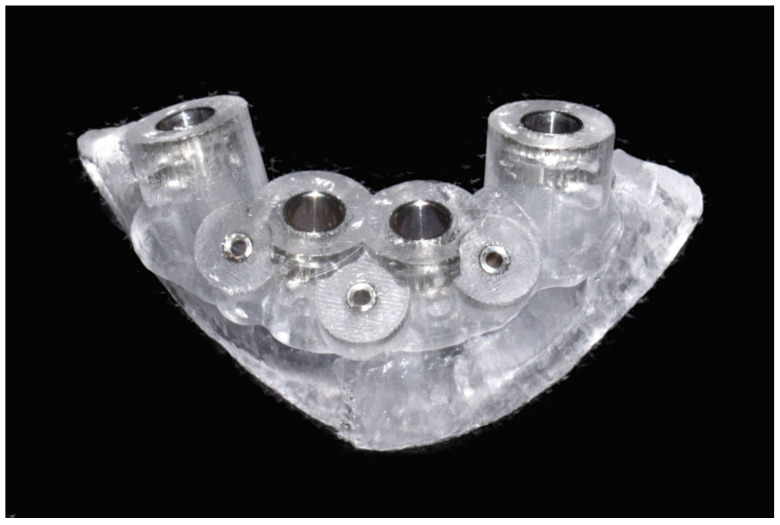
Final stereolithographic surgical drilling guides.

**Figure 35 jpm-13-00129-f035:**
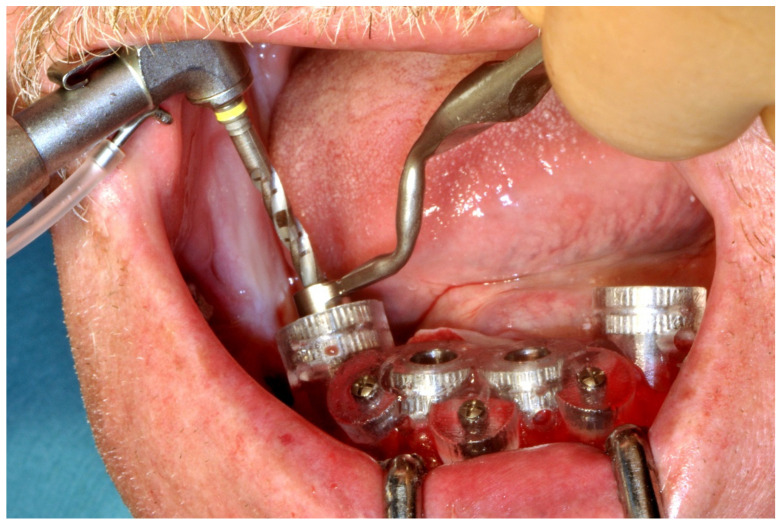
Implant bed preparation by using guided drills inserted into specific ad hoc drill handles.

**Figure 36 jpm-13-00129-f036:**
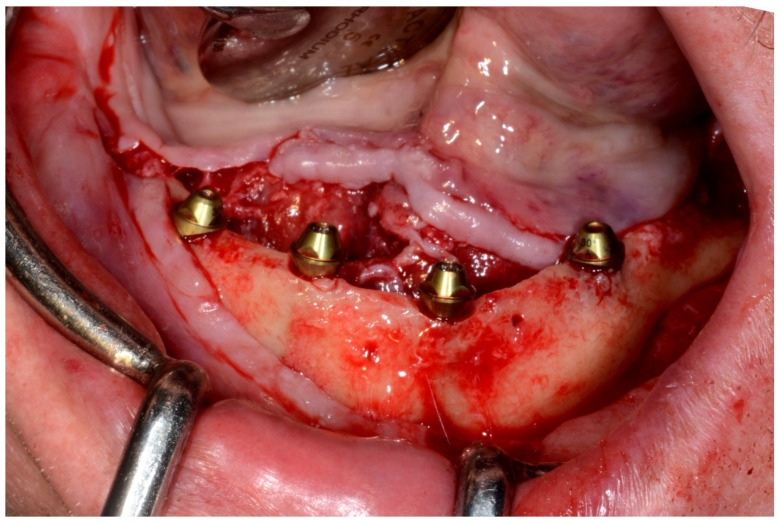
Mandibular implants in place.

**Figure 37 jpm-13-00129-f037:**
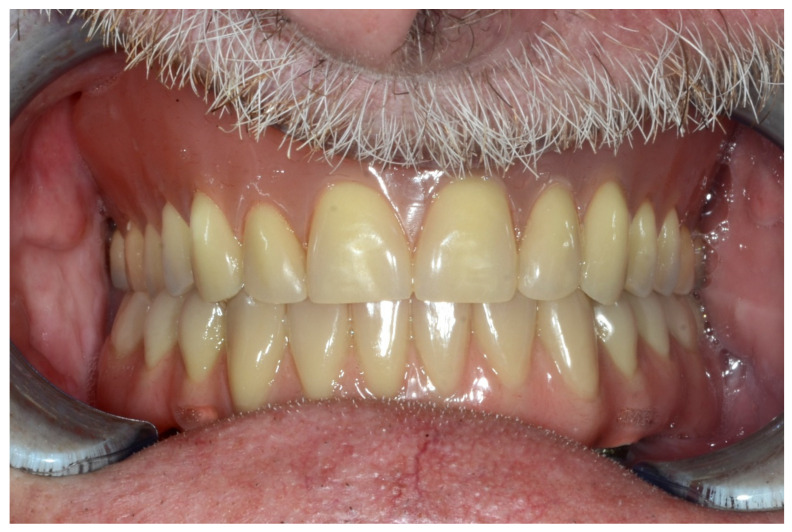
Intraoral view showing the final prosthesis installed.

**Figure 38 jpm-13-00129-f038:**
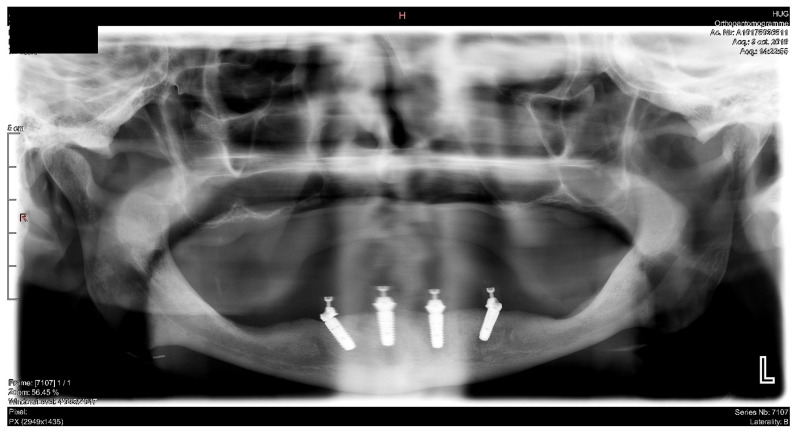
Panoramic radiograph showing the dental implants in place.

**Table 1 jpm-13-00129-t001:** Patient characteristics and clinical data.

Patient No	Gender	Age (Years)	Etiology of Edentulism	Type of Edentulism	Pre-Prosthetic Surgical Procedures	No of Implants	Final Restoration	Follow-Up Period	Complications
1	F	55	DC + PD	Completely edentulous maxilla and mandible	Le Fort I osteotomy with IICG	7 (Mx) 4 (Md)	Bar-retained overdenture	3 years	None
2	F	47	DC + PD	Completely edentulous maxilla	Le Fort I osteotomy with IICG	8 (Mx)	Fixed screw-retained prosthesis	1 year	None
3	F	50	DC + PD	Partially edentulous maxilla	Le Fort I osteotomy with IICG	5 (Mx)	Fixed screw-retained prosthesis	3 years	None
4	F	55	DC + PD	Completely edentulous maxilla and mandible	None	4 (Mx) 4 (Md)	Bar-retained overdenture	3 years	None
5	F	20	AI	Completely edentulous maxilla and mandible	None	8 (Mx) 6 (Md)	Fixed screw-retained prosthesis	2 years	None
6	F	19	AI	Partially edentulous maxilla and complete edentulous mandible	None	6 (Mx) 6 (Md)	Fixed screw-retained prosthesis	2 years	None
7	F	21	ED	Completely edentulous mandible	Bilateral IANLT	6 (Mx)	Fixed screw-retained prosthesis	2 years	None
8	F	67	DC + PD	Partially edentulous mandible	Bilateral IANLT	5 (Mx)	Fixed screw-retained prosthesis	1 year	None
9	F	71	DC + PD	Partially edentulous maxilla and mandible	None	6 (Mx) 4 (Md)	Fixed screw-retained prosthesis	1 year	None
10	F	64	Mandibular resection (MEC)	Partially edentulous mandible	Free fibular graft	3 (Md)	Bar-retained overdenture	3 years	None
11	M	30	Post-trauma	Partially edentulous anterior maxilla	OICG	2 (Mx)	Fixed screw-retained prosthesis	5 years	None
12	M	28	Post-trauma	Partially edentulous anterior maxilla	OICG	3 (Mx)	Fixed screw-retained prosthesis	5 years	None
13	M	74	Mandibular resection (SCC)	Completely edentulous mandible	None	4 (Mx)	Bar-retained overdenture	5 years	None

Abbreviations: F, female; M, male; DC, dental caries; PD, periodontal disease; AI, amelogenesis imperfecta; ED, ectodermal dysplasias; MEC, muco-epidermoid carcinoma; SCC, squamous cell carcinoma; IICG, interpositional iliac crest graft; IANLT, inferior alveolar nerve lateral transposition; OICG, onlay iliac crest graft; Mx, maxilla; Md, mandible.

## Data Availability

The authors declare that the data supporting the findings of this study are available within the paper.

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
