# Peer review of "Computer-Aided Design and Computer-Aided Modeling (CAD/CAM) for Guiding Dental Implant Surgery: Personal Reflection Based on 10 Years of Real-Life Experience"

_jpm, 2023, doi:10.3390/jpm13010129_

Round 1
Reviewer 1 Report
Thank the authors for choosing JPM and MDPI to publish their manuscript.
The use of computer-assisted surgery through CAD/CAM technology is a hot topic in recent decades. The authors perfectly introduce this type of surgery.
regarding materials and methods the authors describe the technique approved by (line 85) "by our local ethical board" refers to the Clinical Ethics Council (CEC) of HUG?
line 100 "in specific Hounsfield units" better describe this prosthetic procedure 186
equipped with separately scanned integrated radiopaque reference spheres with the same CBCT specifications for the patient?
Illustrative cases: for better interpretation of the pictures and also the cases, it is preferable to put the case description first which is put below , and for each picture the description below (Figure 1 A, Figure 1B etc.)
Line 291 "our hospital" change in Hopitaux universitaier ... etc
The discussions are well articulated and comment fully on the references
Author Response
We would like to thank the editor and the reviewers for taking the time to review our manuscript and for the useful comments provided. Please find below and itemized list of responses to reviewer comments, with each individual response found below the original comment of the reviewer.
Review 1
Thank the authors for choosing JPM and MDPI to publish their manuscript.
The use of computer-assisted surgery through CAD/CAM technology is a hot topic in recent decades. The authors perfectly introduce this type of surgery.
Regarding materials and methods the authors describe the technique approved by (line 85) "by our local ethical board" refers to the Clinical Ethics Council (CEC) of HUG?
We thank the reviewer for pointing out this important point. This has been changed in the manuscript accordingly.
Line 100 "in specific Hounsfield units" better describe this prosthetic procedure 186
equipped with separately scanned integrated radiopaque reference spheres with the same CBCT specifications for the patient?
We thank the reviewer for having raised this important point, and we apologize for not having clarified this aspect. The explanation of the procedure has been rephrased.
Illustrative cases: for better interpretation of the pictures and also the cases, it is preferable to put the case description first which is put below , and for each picture the description below (Figure 1 A, Figure 1B etc.)
As suggested, this has been changed in the manuscript accordingly.
Line 291 "our hospital" change in Hopitaux universitaier ... etc
This has been changed in the manuscript accordingly.
The discussions are well articulated and comment fully on the references
Reviewer 2 Report
Dear Editors,
The article entitled "Computer-aided design and computer-aided manufacturing (CAD /CAM) for guiding dental implant surgery: Reflection based on 10 years of real-life experience" describes very well and methodically the specifically surgical, implantological course of computer-aided surgery (CAS) approaches in medium and extremely severe clinical cases of complete and partial edentulism. A very useful read for clinicians who are or will be dealing with this issue. I have two suggestions for authors. If it is possible for all 3 clinical cases, the authors should show the control OPGs after the dental implant insertion and the control OPGs after the mentioned wearing periods of the prosthetic restorations (5 years; 8 years and 3 years). I do not insist on this, but it seems to me that the presentation of the clinical cases would be more consistent regarding the level of marginal bone resorption around the dental implants. At the end of the discussion, lines 380 - 383, I suggest that the sentence mentioning the cost prices of the CAD /CAM surgical guides be omitted or reworded so that the prices are not mentioned (there has been a price reduction).
Author Response
We would like to thank the editor and the reviewers for taking the time to review our manuscript and for the useful comments provided. Please find below and itemized list of responses to reviewer comments, with each individual response found below the original comment of the reviewer.
Review 2
Dear Editors,
The article entitled "Computer-aided design and computer-aided manufacturing (CAD /CAM) for guiding dental implant surgery: Reflection based on 10 years of real-life experience" describes very well and methodically the specifically surgical, implantological course of computer-aided surgery (CAS) approaches in medium and extremely severe clinical cases of complete and partial edentulism. A very useful read for clinicians who are or will be dealing with this issue. I have two suggestions for authors. If it is possible for all 3 clinical cases, the authors should show the control OPGs after the dental implant insertion and the control OPGs after the mentioned wearing periods of the prosthetic restorations (5 years; 8 years and 3 years). I do not insist on this, but it seems to me that the presentation of the clinical cases would be more consistent regarding the level of marginal bone resorption around the dental implants.
We apologize but unfortunately the required documents are not available.
At the end of the discussion, lines 380 - 383, I suggest that the sentence mentioning the cost prices of the CAD /CAM surgical guides be omitted or reworded so that the prices are not mentioned (there has been a price reduction).
This has been changed in the manuscript accordingly.